# Exploring Behavior-Relevant and Disentangled Neural Dynamics with Generative Diffusion Models

**Yule Wang**
Georgia Institute of Technology
Atlanta, GA, 30332 USA
yulewang@gatech.edu

**Chengrui Li**
Georgia Institute of Technology
Atlanta, GA, 30332 USA
cnlichengrui@gatech.edu

**Weihan Li**
Georgia Institute of Technology
Atlanta, GA, 30332 USA
weihanli@gatech.edu

**Anqi Wu**
Georgia Institute of Technology
Atlanta, GA, 30332 USA
anqiwu@gatech.edu

## Abstract

Understanding the neural basis of behavior is a fundamental goal in neuroscience. Current research in large-scale neuro-behavioral data analysis often relies on decoding models, which quantify behavioral information in neural data but lack details on behavior encoding. This raises an intriguing scientific question: "*how can we enable in-depth exploration of neural representations in behavioral tasks, revealing interpretable neural dynamics associated with behaviors*". However, addressing this issue is challenging due to the varied behavioral encoding across different brain regions and mixed selectivity at the population level. To tackle this limitation, our approach, named "BeNeDiff", first identifies a fine-grained and disentangled neural subspace using a behavior-informed latent variable model. It then employs state-of-the-art generative diffusion models to synthesize behavior videos that interpret the neural dynamics of each latent factor. We validate the method on multi-session datasets containing wide-field calcium imaging recordings across multiple brain regions of the dorsal cortex. By guiding the diffusion model to activate individual latent factors, we verify that the neural dynamics of latent factors in the disentangled neural subspace provide interpretable quantifications of the behaviors of interest across multiple brain regions. Meanwhile, the neural subspace in BeNeDiff demonstrates high disentanglement and neural reconstruction quality. Our codes are available at https://github.com/BRAINML-GT/BeNeDiff.

## 1 Introduction

Understanding and elucidating the complex interrelationships between behavioral data and neural population activity is a long-standing goal in systems neuroscience [Batty et al., 2019; Gomez-Marin et al., 2014; Krakauer et al., 2017; Berman, 2018]. Exploring the neural basis of behavior not only deepens our basic knowledge of brain functions but also establishes a foundation for developing improved treatments for psychiatric and neurological conditions [Vieira et al., 2017; Ibáñez et al., 2018]. Significant progress has been achieved in developing computational toolkits for neuro-behavioral decoding by using behavior video data [Whiteway et al., 2021; Batty et al., 2019; Musall et al., 2019; Stringer et al., 2019]. These methods perform region-based behavior decoding to map neural activity across multiple brain regions of the dorsal cortex to the behaviors from the videos. However, these methods only quantify how much behavioral information is encoded in neural populations, but do not reveal the details of such encoding. There has been markedly less focus,

38th Conference on Neural Information Processing Systems (NeurIPS 2024).

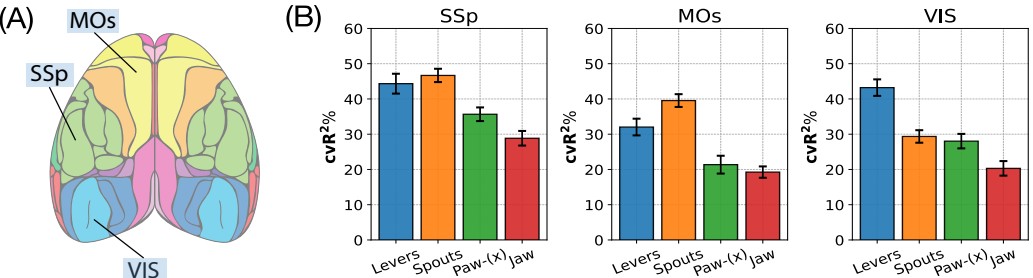

Figure 1: **Empirical study across multiple brain regions of dorsal cortex neural recordings of a mouse in a visual decision-making task**. **(A)** The Brain Atlas map [Lein et al., 2007]. **(B)** Neural signals in various brain regions (SSp, MOs, and VIS) exhibit mixed selectivity in behavior of interest decoding. "Levers", "Spouts", "Paw-(x)", and "Jaw" are four behaviors of interest. $\mathbf{cvR}^2$ is short for cross-validation coefficient of determination. The higher, the better.

with cortex-wide signals, on enabling in-depth exploration of neural activities during behavioral tasks, where specific neural patterns reveal dynamic evolutions corresponding to distinct behaviors of interest. However, empirically addressing this scientific question is challenging due to neural population activities in various brain regions exhibiting mixed selectivity [Sani et al., 2021; Hasnain et al., 2023], responding robustly to multiple behaviors of interest. We further verify this finding through an empirical study across three brain regions on the dorsal cortex of head-fixed mice [Musall et al., 2019] (shown in Figure 1).

To tackle this issue, we propose a method - Exploring **Be**havior-Relevant and Interpretable **Neu**ral Dynamics with Generative **Diff**usion Models - ("BeNeDiff"). We first employ a neural latent variable model (LVM) to identify orthogonal and disentangled neural latent subspace. This is achieved through a semi-supervised variational autoencoder, which integrates behavioral labels to rotate the subspace. Subsequently, our main idea is to explore the neural dynamics of each latent factor in the learned subspace for distinct quantifications of the behaviors of interest. However, such a workflow is non-trivial since naïve latent manipulation produces samples not conform to the original distribution, leading to mapped video-based behavioral data that loses its validity (we further detail this part in Method Section 3.2.1).

Notably, we aim to investigate the behavioral-specificity of neural latent factors in a generative fashion. We leverage state-of-the-art video diffusion models (VDMs) to generate behavior videos predicted to *activate* individual latent factors along the single-trial trajectory. Technically, the VDMs are capable of capturing the overall temporal dynamics and synthesizing behavior videos in a classifier-guided manner [Dhariwal and Nichol, 2021]. Inspired by Noise-Contrastive Estimation [Gutmann and Hyvärinen, 2010], the guidance objective is formulated to amplify the variance of the selected latent factor along its neural trajectory while suppressing the variance of the neural trajectories of the other latent factors.

We conduct experiments to verify the efficacy of BeNeDiff on a widefield calcium imaging dataset, where a head-fixed mouse performs a visual decision-making task across multiple sessions [Musall et al., 2018; 2019]. The neural subspace in BeNeDiff exhibits high levels of disentanglement and neural reconstruction quality, as evidenced by multiple quantitative metrics. By guiding the diffusion model to activate individual latent factors, we verify that the neural dynamics within the disentangled subspace provide interpretable and selective quantifications of the behaviors of interest (e.g., paw movements) across multiple brain regions. These results advance our understanding of neuro-behavioral relationships through the identification of fine-grained behavioral subspaces and the uncovering of disentangled neural dynamics.

To highlight our major contributions: (1) This is the first work to explore wide-field imaging across multiple brain regions of the dorsal cortex of head-fixed mice during a decision-making task using neural subspace analysis, rather than merely performing neuro-behavior decoding. We uncover disentangled neural representations for various behaviors. (2) To visualize the behavior dynamics within a disentangled neural subspace of each brain region, we develop a novel VDM-based interpretation tool that faithfully reflects behavior-related neural dynamics. It is essential to interpret the meaning of each neural latent dimension as well as the behavior dynamics it encodes.

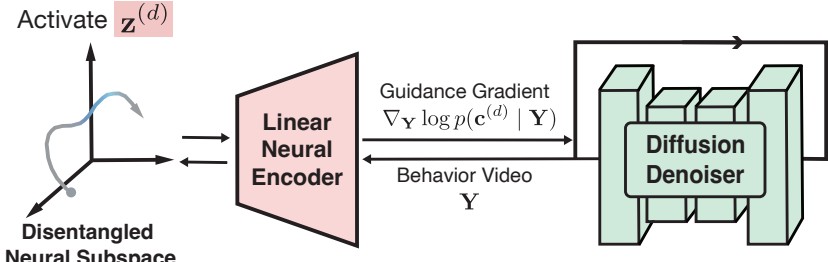

Figure 2: **Schematic diagram of neural dynamics interpretation with BeNeDiff**. We first employ a neural LVM to identify a disentangled neural latent subspace (the left part). Then, we train a linear neural encoder to map behavior video frames to neural trajectories. We use video diffusion models (VDMs) to generate behavior videos guided by the neural encoder, based on the objective of *activating the variance* of individual latent factors along the single-trial trajectory. This approach provides interpretable quantifications of neural dynamics in relation to the behaviors of interest.

## 2 Preliminaries

### 2.1 Problem Formulation

We first provide the notations of the paired neuro-behavioral observations. The single-trial neural population activities are denoted as $\mathbf{X} = [\mathbf{x}_1, \ldots, \mathbf{x}_L]^\top \in \mathbb{R}^{L \times N}$, where $L$ is the trial length (*i.e.*, number of time bins), $N$ is the number of observed neural signals. The behavioral video frames are denoted as $\mathbf{Y} = [\mathbf{Y}_1, \ldots, \mathbf{Y}_L]^\top \in \mathbb{R}^{L \times H \times W}$, where $H$, $W$ are the height and width of the compressed behavior video frames. We extract behavior labels $\mathbf{U} = [\mathbf{u}_1, \ldots, \mathbf{u}_L]^\top \in \mathbb{R}^{L \times B}$ from the video frames using a behavior LVM [Whiteway et al., 2021]. $B$ is the number of the behavior.

We build a variational autoencoder (VAE) [Kingma and Welling, 2013] to infer the neural latent trajectories $\mathbf{Z} = [\mathbf{z}_1, \ldots, \mathbf{z}_L]^\top \in \mathbb{R}^{L \times D}$, which are also informed by behavioral labels. $D$ is the latent factor number. We denote its probabilistic encoder and decoder as $q_{\boldsymbol{\psi}}(\mathbf{Z} \mid \mathbf{X}, \mathbf{U})$ and $p_{\boldsymbol{\phi}}(\mathbf{X}, \mathbf{U} \mid \mathbf{Z})$, respectively. We denote the neural trajectory of a single latent factor as $\mathbf{z}^{(d)} = \mathbf{Z}_{:,d}$, where $d \in \{1, 2, \cdots, D\}$. Our primary goal is to investigate the neural dynamics of $\mathbf{z}^{(d)}$ through selectivity quantifications of its corresponding single-trial behavioral video data $\mathbf{Y}$.

### 2.2 Generative Video Diffusion Models

Diffusion models have also achieved impressive results in video synthesis over recent years [Ho et al., 2022b;a; Harvey et al., 2022]. VDMs process a fixed number of frames and factorize them over the temporal dimension via a deep neural network [Ho et al., 2022a; Harvey et al., 2022]. The training of VDMs starts from a forward process with a variance schedule $\{\beta_1, \ldots, \beta_T\}$, the noised sample $\mathbf{Y}_t$ follows the Gaussian conditional: $q(\mathbf{Y}_t \mid \mathbf{Y}_0) := \mathcal{N}(\mathbf{Y}_t; \sqrt{\bar{\alpha}_t}\mathbf{Y}_0, (1 - \bar{\alpha}_t)\mathbf{I})$, where $\alpha_t := 1 - \beta_t$ and $\bar{\alpha}_t := \prod_{s=1}^{t} \alpha_s$. A denoising model $\hat{\epsilon}_\theta(\cdot)$ is trained to reverse the forward process using a weighted mean squared error loss:

$$\mathcal{L}_{\text{VDM}}(\boldsymbol{\theta}) = \mathbb{E}_{\boldsymbol{\epsilon} \sim \mathcal{N}(\mathbf{0}, \mathbf{I})}, \mathbb{E}_{t \sim \mathcal{U}[0,T]} \left[ w(\lambda_t) \|\boldsymbol{\epsilon} - \hat{\boldsymbol{\epsilon}}_{\boldsymbol{\theta}}(\mathbf{Y}_t, t)\|_2^2 \right], \tag{1}$$

in which time-steps $t$ are uniformly sampled and $w(\lambda_t)$ is the weighting ratio. This loss function can be justified as optimizing a weighted variational lower bound on the data log-likelihood. In the sampling phase, we start from $\mathbf{Y}_T \sim \mathcal{N}(\mathbf{0}, \mathbf{I}_{L \times H \times W})$ and perform step-by-step denoising,

$$\mathbf{Y}_{t-1} = \frac{1}{\sqrt{\alpha_t}} \left( \mathbf{Y}_t - \frac{1 - \alpha_t}{\sqrt{1 - \bar{\alpha}_t}} \hat{\boldsymbol{\epsilon}}_{\boldsymbol{\theta}}(\mathbf{Y}_t, t) \right) + \sigma_t \boldsymbol{\epsilon}_t, \tag{2}$$

where random noise perturbation $\boldsymbol{\epsilon}_t \sim \mathcal{N}(\mathbf{0}, \mathbf{I}_{L \times H \times W})$ for timesteps $t > 1$, $\boldsymbol{\epsilon}_t = \mathbf{0}$ when $t = 1$, and $\sigma_t^2 = \frac{1 - \bar{\alpha}_{t-1}}{1 - \bar{\alpha}_t} \beta_t$.

## 3 Methods

Then, we train a linear neural encoder from the behavior video frames to the neural trajectories. We leverage video diffusion models (VDMs) to generate behavior videos guided by the neural encoder,

based on the objective of *activating the variance* of individual latent factors along the single-trial trajectory, providing interpretable quantifications of neural dynamics with respect to the behaviors of interest.

In this section, we first detail the process by which BeNeDiff infers a disentangled neural latent subspace. We then discuss the approach that BeNeDiff interprets the selectivity of neural dynamics of latent factors using the video diffusion model.

## 3.1 Behavior-Relevant and Disentangled Neural Latent Subspace Learning

Drawing inspiration from recent progress in the field of neural LVMs [Kingma et al., 2014; Klys et al., 2018], we employ a VAE to learn a disentangled neural subspace. The neural data $\mathbf{X}$ usually contains a good amount of information other than behavior [Hasnain et al., 2023], thus an unsupervised disentangled VAE won't effectively discover disentangled subspace with behavior only. Therefore, we introduce behavior labels $\mathbf{U}$ to inform the VAE to learn a latent subspace that better accounts for the variance related to behavior. We note that this technique is widely adopted in previous neuro-behavioral analysis works [Wang et al., 2024; Schneider et al., 2023; Gondur et al., 2023]. Notably, to enforce the disentanglement in the latent subspace, we incorporate a *total-correlation* (TC) penalty term [Chen et al., 2018] to enforce the VAE to find statistically independent latent factors in the semi-supervised setting. The VAE optimizes the following evidence lower bound (ELBO) [MacKay, 2003]:

$$
\log p_{\boldsymbol{\phi}}(\mathbf{X}, \mathbf{U}) \geq \mathbb{E}_{q_{\boldsymbol{\psi}}(\mathbf{Z}|\mathbf{X},\mathbf{U})}\Big[ \underbrace{\log p_{\boldsymbol{\phi}}(\mathbf{X} \mid \mathbf{Z})}_{\text{Neural Reconstruction}} + \underbrace{\log p_{\boldsymbol{\phi}}(\mathbf{U} \mid \mathbf{Z})}_{\text{Behavior Info.}} \Big] - \underbrace{\mathbb{D}_{\mathrm{KL}}\Big( q_{\boldsymbol{\psi}}(\mathbf{Z} \mid \mathbf{X}, \mathbf{U}) \big\| p(\mathbf{Z}) \Big)}_{\text{Regularization Term}}
$$

$$
- \beta \underbrace{\mathbb{D}_{\mathrm{KL}}\Big( q_{\boldsymbol{\psi}}(\mathbf{Z} \mid \mathbf{X}, \mathbf{U}) \big\| \prod_d q_{\boldsymbol{\psi}}(\mathbf{z}^{(d)} \mid \mathbf{X}, \mathbf{U}) \Big)}_{\text{Total Correlation}} =: -\mathcal{L}_{\mathrm{VAE}}(\boldsymbol{\phi}, \boldsymbol{\psi})
$$

(3)

in which $\mathbf{z}^{(d)}$ denotes the neural trajectory of the $d$-th latent factor, and the value of $\beta$ controls the strength of disentanglement penalty. However, the factorial density in this term is untractable in practice, so here we use the minibatch-weighted sampling estimator [Chen et al., 2018] to approximate the TC penalty term. We note that the variational autoencoder employs a sequential architecture [Fabius and Van Amersfoort, 2014] to capture the overall temporal dynamics along the single-trial trajectory $\{\mathbf{x}_l\}_{l=1}^L$, plugging bi-directional recurrent units [Schuster and Paliwal, 1997] into both the probabilistic encoder $q_{\boldsymbol{\psi}}(\cdot)$ and decoder $p_{\boldsymbol{\phi}}(\cdot)$.

## 3.2 Diffusion Guided Video Generation for Neural Dynamics Interpretation

### 3.2.1 Downside of Latent Manipulation for Interpreting Neural Dynamics

As for testifying the neural dynamics of a single disentangled latent factor $\mathbf{z}^{(d)}$ on the behavioral videos $\mathbf{Y}$, a straightforward attempt is to train a neural-net model to approximate the posterior distribution $p(\mathbf{Y} \mid \mathbf{Z})$ and then perform latent manipulation on each single latent factor. There are two major techniques to perform latent manipulation. The first is a naïve manipulation. This method manipulates a single subspace $\mathbf{z}^{(d)}$ while keeping the non-target latent factors fixed at arbitrary values. It then observes how the manipulation affects $\mathbf{Y}$. The induced changes in the videos reveal the dynamics encoded by $\mathbf{z}^{(d)}$. The second method uses classifier-free guidance [Ho and Salimans, 2022], where we allow the activated latent factor $\mathbf{z}^{(d)}$ to evolve while fixing non-target latent factors to arbitrary values. However, setting arbitrary values without knowing the true distributions of non-target subspaces can lead to unnatural distortions in generated videos, complicating the visualization and interpretation of genuine animal behavioral dynamics.

### 3.2.2 Behavioral Video Generation for Neural Dynamics Interpretation

So here we employ the video diffusion models (VDMs) to explore factor-wise neural dynamics through a generative manner, which is capable of maintaining temporal consistency and behavioral dynamics across frames. The primary goal is to perform behavior data generation conditioned on activating a single latent factor along the neural trajectory. Thus the resulting behavior video can

provide interpretable quantifications of the neural dynamics of factor $\mathbf{z}^{(d)}$. Specifically, we implement classifier guidance [Kawar et al., 2022]. By Bayes rule, we obtain the following posterior density and gradient [Mardani et al., 2023]:

$$p_{\boldsymbol{\theta},\boldsymbol{\lambda}}\left(\mathbf{Y}_t \mid \mathbf{c}\right) = p_{\boldsymbol{\theta}}(\mathbf{Y}_t)p_{\boldsymbol{\lambda}}\left(\mathbf{c} \mid \mathbf{Y}_t\right)/p(\mathbf{c}), \tag{4}$$

$$\nabla_{\mathbf{Y}_t}\log p_{\boldsymbol{\theta},\boldsymbol{\lambda}}\left(\mathbf{Y}_t \mid \mathbf{c}\right) = \underbrace{\nabla_{\mathbf{Y}_t}\log p_{\boldsymbol{\theta}}(\mathbf{Y}_t)}_{\text{Unconditional Gradient}} + \underbrace{\nabla_{\mathbf{Y}_t}\log p_{\boldsymbol{\lambda}}\left(\mathbf{c} \mid \mathbf{Y}_t\right)}_{\text{Guidance Gradient}}, \tag{5}$$

in which $\boldsymbol{\theta}, \boldsymbol{\lambda}$ are the parameter sets for the classifier and the denoising model, respectively. Note that $t$ indicates the time step in the diffusion model. Our goal is to estimate the two terms on the RHS of Eq. (5) to perform conditional denoising in each step. We first approximate the density of the behavior video data through a standard denoising model $\hat{\boldsymbol{\epsilon}}_{\theta}\left(\mathbf{Y}_t, t\right)$ according to Eq. (1) since the first unconditional gradient term can be derived through it:

$$\nabla_{\mathbf{Y}_t}\log p_{\boldsymbol{\theta}}\left(\mathbf{Y}_t\right) = -\frac{1}{\sqrt{1-\bar{\alpha}_t}}\hat{\boldsymbol{\epsilon}}_{\theta}\left(\mathbf{Y}_t, t\right). \tag{6}$$

For the calculation of the guidance term, we first train a linear neural encoder as the classifier from the behavior video data to the neural latent variables of the learned semi-supervised VAE subspace. We denote the estimated neural latent trajectories as $\hat{\mathbf{Z}}_t = [\hat{\mathbf{z}}_{t,1}, \ldots, \hat{\mathbf{z}}_{t,L}]^{\top} \in \mathbb{R}^{L \times D}$, in which:

$$\hat{\mathbf{z}}_{t,l} = \mathbf{W}\,\text{vec}(\mathbf{Y}_{t,l}) + \mathbf{q}; \quad \mathbf{q} \sim \mathcal{N}\left(\mathbf{0}, \mathbf{Q}\right), \tag{7}$$

where $1 \leq l \leq L$, $\hat{\mathbf{z}}_{t,l} \in \mathbb{R}^D$ denotes the estimated value of latent factors at time bin $l$ and diffusion step $t$. The parameter set of the linear encoder $\boldsymbol{\lambda} = \{\mathbf{W}, \mathbf{Q}\}$. $\mathbf{W} \in \mathbb{R}^{D \times M}$ is the linear transformation matrix, $\mathbf{Q} \in \mathbb{R}^{D \times D}$ is the covariance matrix and $\text{vec}(\cdot)$ represents vectorizing the two-dimensional video frame into column vector. After training the encoder, we fix all parameters and use it to construct the density $p_{\boldsymbol{\lambda}}\left(\mathbf{c} \mid \mathbf{Y}_t\right)$.

The class labels $\mathbf{c} \in \left\{\mathbf{c}^{(1)}, \mathbf{c}^{(2)}, \ldots, \mathbf{c}^{(D)}\right\}$, in which $\mathbf{c}^{(d)}$ is a one-hot column vector with a one at the $d$-th dimension and zeros elsewhere. Drawing inspiration from Noise-Contrastive Estimation [Gutmann and Hyvärinen, 2010], our guidance objective of the activation of latent factor $d$-th is formulated as maximizing the variance of the trajectory $\hat{\mathbf{z}}^{(d)}$ while minimizing the variance of the other latent factor trajectories in $\hat{\mathbf{Z}}$:

$$\log p_{\boldsymbol{\lambda}}\left(\mathbf{c}^{(d)}\Big|\mathbf{Y}_t\right) = \log\left[\frac{\exp\left(f_{\boldsymbol{\lambda}}^{+}\left(\hat{\mathbf{Z}}, \mathbf{c}^{(d)}\right)/\tau\right)}{\exp\left(f_{\boldsymbol{\lambda}}^{+}\left(\hat{\mathbf{Z}}, \mathbf{c}^{(d)}\right)/\tau\right) + \sum_{k=1}^{K}\exp\left(f_{\boldsymbol{\lambda}}^{-}\left(\hat{\mathbf{Z}}, \mathbf{c}^{(d)}\right)/\tau\right)}\right], \tag{8}$$

where $f_{\boldsymbol{\lambda}}^{+}\left(\hat{\mathbf{Z}}, \mathbf{c}^{(d)}\right) = \text{Var}\left(\hat{\mathbf{Z}}\right)\mathbf{c}^{(d)}$ calculates the variance of the selected latent factor and $f_{\boldsymbol{\lambda}}^{-}\left(\hat{\mathbf{Z}}, \mathbf{c}^{(d)}\right) = \text{Var}\left(\hat{\mathbf{Z}}\right)\mathbf{c}^{(j)}, \; j \sim \text{Uniform}(\{1, 2, \ldots, D\}\backslash\{d\})$ calculates the variance of another sampled latent factor's trajectory. $\text{Var}\left(\hat{\mathbf{Z}}\right) \in \mathbb{R}^{1 \times D}$ is a row vector where each element is the variance of every latent factor along the neural trajectory. $\tau$ is the temperature parameter. $K$ is a hyperparameter controlling the number of sampled negative samples at each iteration.

The gradient $\nabla_{\mathbf{Y}_t}\log p_{\boldsymbol{\lambda}}\left(\mathbf{c}^{(d)} \mid \mathbf{Y}_t\right)$ is computed using automatic differentiation [Paszke et al., 2017]. Algorithm 1 describes the guided behavior video generation steps of our proposed framework BeNeDiff.

---

**Algorithm 1:** Generative Video Diffusion Model for Neural Dynamics Interpretation

---

**Input:** Condition label $\mathbf{c}^{(d)}$ for interpreting the neural dynamics of the $d$-th latent factor
Initiate $\mathbf{Y}_T \sim \mathcal{N}(\mathbf{0}, \mathbf{I}_{L \times H \times W})$
**for** $t = T$ **to** 1 **do**
$\quad \hat{\boldsymbol{\epsilon}}_{\boldsymbol{\theta},\boldsymbol{\lambda}}'\left(\mathbf{Y}_t, t\right) = \hat{\boldsymbol{\epsilon}}_{\boldsymbol{\theta}}\left(\mathbf{Y}_t, t\right) - \sqrt{1-\bar{\alpha}_t}\nabla_{\mathbf{Y}_t}\log p_{\boldsymbol{\lambda}}\left(\mathbf{c}^{(d)} \mid \mathbf{Y}_t\right)$
$\quad \boldsymbol{\epsilon}_t \sim \mathcal{N}(\mathbf{0}, \mathbf{I})$ if $t > 1$, else $\boldsymbol{\epsilon}_t = \mathbf{0}$
$\quad \mathbf{Y}_{t-1} = \frac{1}{\sqrt{\alpha_t}}\left(\mathbf{Y}_t - \frac{1-\alpha_t}{\sqrt{1-\bar{\alpha}_t}}\hat{\boldsymbol{\epsilon}}_{\boldsymbol{\theta},\boldsymbol{\lambda}}'\left(\mathbf{Y}_t, t\right)\right) + \sigma_t\boldsymbol{\epsilon}_t$
**end**
**Output:** Generated behavior video $\mathbf{Y}_0$

---

# 4    Related Works

**Disentangled Latent Subspace Learning.**  Neural LVMs is a fundamental framework which posits that single-trial neural population activities rely on low-dimensional "neural manifolds" [Gallego et al., 2018; Mitchell-Heggs et al., 2023; Li et al., 2023a; Hurwitz et al., 2021] and their extracted latent variables are successful in describing single-trial neural activities [Li et al., 2024c; 2022; 2024d; Liu et al., 2021; 2022; Li et al., 2024a]. Learning disentangled latent variables that uncover statistically independent latent factors [Chen et al., 2018] can provide enhanced robustness, interpretability, and controllability. Typically, this type of work involves adding auxiliary regularizer terms to enhance orthogonality [Mathieu et al., 2019] and reduce the total correlation [Chen et al., 2018] among the latent factors. In neuroscience, there have been studies focusing on the disentanglement of latent subspace within rich behavioral data [Whiteway et al., 2021; Shi et al., 2021]. However, our work is the first to discover interpretable and disentangled latent subspaces of wide-field imaging data.

**Generative Diffusion Models.**  In recent years, diffusion models have achieved great success in generating high-quality images due to their expressivity and flexibility [Ho et al., 2020; Song et al., 2020a;b; Vahdat et al., 2021]. Moreover, for the more challenging task of video generation, there have been several explorations using diffusion models to address it. From a modeling perspective, the key concern is how to maintain temporal dynamics and consistency across frames. Most existing works [Ho et al., 2022b;a] extend the 2D U-Net architecture [Ronneberger et al., 2015; Song et al., 2024] to a 3D framework by considering the time axis. In this 3D framework, convolutions are performed in both spatial and temporal dimensions. Additionally, recent studies in neural computation have leveraged generative diffusion models to tackle domain-specific tasks, such as neural distribution alignment [Wang et al., 2024] and decoding visual stimulus from brain activities [Sun et al., 2024b;a]. Our work is the first to employ generative diffusion models for analyzing neuro-behavioral data relationships.

# 5    Experimental Results

## 5.1    Dataset Description

A head-fixed mouse performed a visual decision-making task while neural activity across the dorsal cortex was optically recorded using widefield calcium imaging [Musall et al., 2019; Churchland et al., 2019]. The mouse's behavior included both instructed and uninstructed movements. For behavioral data acquisition, two cameras captured video frames from both a side view and a bottom view. The dataset comprises 1126 trials conducted over two sessions, with 189 frames per trial at a frame rate of 30 Hz. Concurrently, neural activity was recorded at the same frame rate. The grayscale video frames were down-sampled to 128×128 pixels. We extract 275 dimensions of neural signals from the high-dimensional widefield imaging data using the open-sourced LocaNMF decomposition toolkit [Saxena et al., 2020]. As shown in Figure 3, the behaviors of interest include the moving lick spouts, moving levers, the single visible right paw trajectories, and the movement of the jaw and chest, all tracked using DeepLabCut [Mathis et al., 2018].

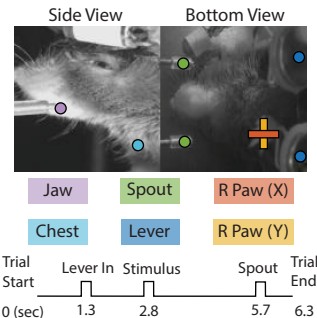

Figure 3:  **Widefield Calcium Imaging Dataset**.  The head-fixed mouse is performing a visual decision-making task, with the behaviors of interest and the trial structure illustrated.

## 5.2    Disentangled Neural Latent Subspace Investigation

We note that we train a unique neural LVM for each individual brain region (single-region), and we evaluate both the behavior decoding and neural reconstruction performance of each brain region-specific neural latent trajectories.

**Single Latent Factor Behavior Decoding.**  In order to verify the disentanglement of the learned neural subspace in BeNeDiff, we evaluate the behavior label decoding performance of each individual latent factor. Specifically, we train a unique linear regressor for each latent factor from the VAE and plot the decoding accuracy as the R-squared value ($R^2\%$). The results of VIS-Right region (the right visual region) are shown in Figure 4. The main observation is that each latent factor is specific to a

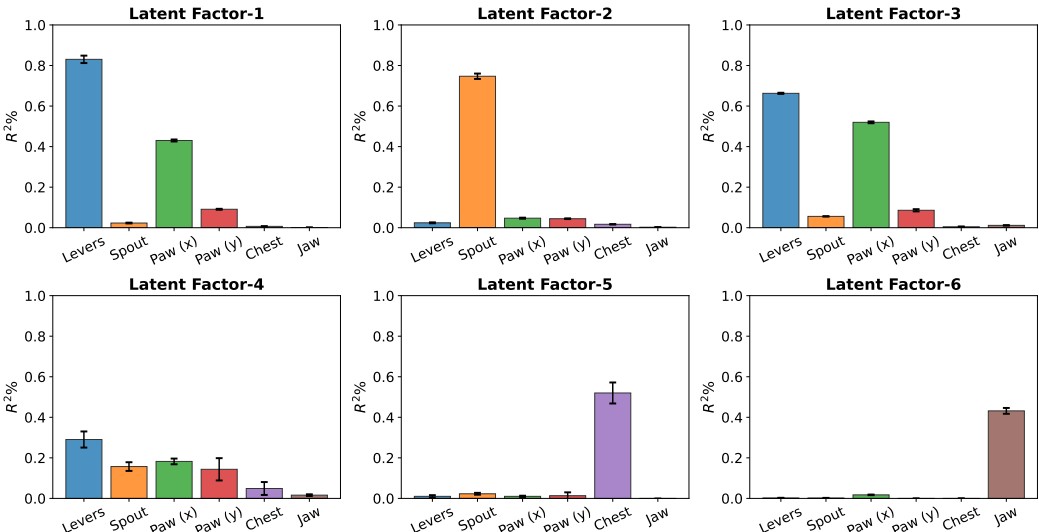

Figure 4: **Behavior decoding results of the disentangled neural latent variables of the VIS-Right region**. We observe that the decoding capability of each latent factor is specified to the corresponding behavior of interest, exhibiting a single-mode shape. In contrast, the original neural signals exhibit mixed selectivity to the behaviors, shown in Figure 1(B). Each experiment condition is repeated 5 times, with the mean represented by the bar plot and the standard deviations shown as error bars.

unique behavior of interest, confirming the orthogonality and clear disentanglement of the inferred latent trajectories from a quantitative perspective.

**Neural Observation Signals Reconstruction.** To prevent the VAE from overfitting to the behavioral labels, BeNeDiff also aims to maintain a low reconstruction error for neural activity. Table 1 presents the quantitative reconstruction results compared to baseline methods, including Semi-Supervised Learning (SSL) [Kingma et al., 2014], CEBRA [Schneider et al., 2023], and pi-VAE [Zhou and Wei, 2020]. The table records the R-squared values ($R^2$, in %) and RMSE for each method. Additionally, we plot the ground-truth neural signals and the reconstructed signals of several methods in a single trial in Figure 5. The main observation is that the neural reconstruction is well-preserved given the behavioral priors. One possible explanation is that the behavioral labels rotate the latent subspace while preserving the necessary information for reconstructing the neural data. The neural signals can be hardly recovered from the behavior labels only. It indicates that the behavior-informed latent does encode significant neural information that is not contained in the behavior labels. Furthermore, we evaluate the disentanglement quality of the latent subspace using the widely-adopted MIG (Mutual Information Gap) metric [Chen et al., 2018], also listed in Table 1. We observe that the learned latent subspace of BeNeDiff significantly enhances disentanglement compared to the vanilla VAE.

Table 1: **Baseline Comparison** of the neural LVM on two brain regions of Session-1. The boldface denotes the highest score of the MIG metric. Each experiment condition is repeated with 5 runs, and their mean and standard deviations are listed.

| Region | Metrics | SSL | CEBRA | pi-VAE | **Ours** |
|---|---|---|---|---|---|
| VIS-Left | $R^2(\%)\uparrow$ | 81.10 ($\pm$0.26) | 79.60 ($\pm$0.22) | 74.37 ($\pm$0.24) | 75.41 ($\pm$0.24) |
| | RMSE $\downarrow$ | 32.77 ($\pm$0.17) | 33.07 ($\pm$0.18) | 36.74 ($\pm$0.22) | 35.50 ($\pm$0.17) |
| | MIG(%) $\uparrow$ | 37.50 ($\pm$0.20) | 40.12 ($\pm$0.24) | 43.98 ($\pm$0.29) | **55.87 ($\pm$0.26)** |
| MOs-Left | $R^2(\%)\uparrow$ | 76.65 ($\pm$0.30) | 72.63 ($\pm$0.28) | 70.73 ($\pm$0.23) | 69.59 ($\pm$0.22) |
| | RMSE $\downarrow$ | 30.64 ($\pm$0.21) | 32.14 ($\pm$0.17) | 35.69 ($\pm$0.19) | 36.91 ($\pm$0.18) |
| | MIG(%) $\uparrow$ | 36.89 ($\pm$0.23) | 37.94 ($\pm$0.23) | 42.20 ($\pm$0.28) | **58.56 ($\pm$0.29)** |

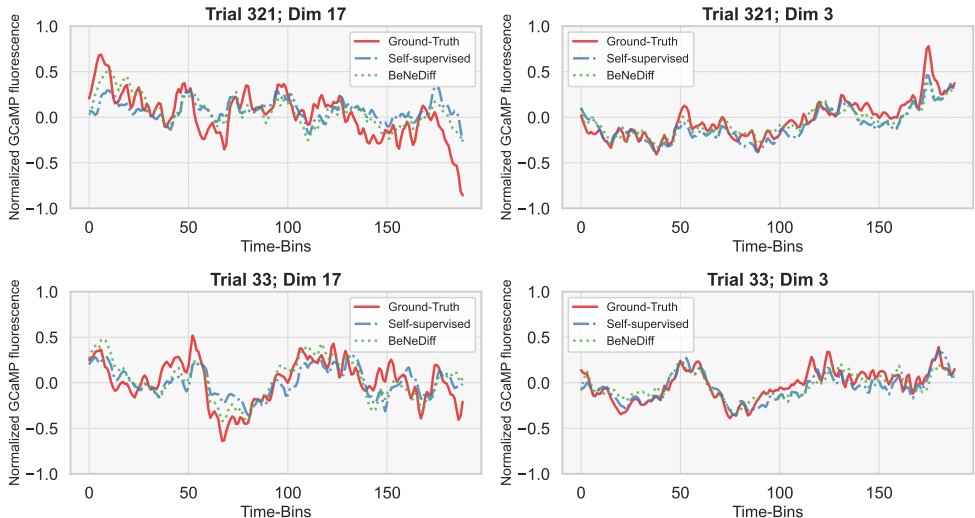

Figure 5: **Neural signal reconstruction performance evaluation of the VIS-Right region**. We observe that the neural reconstruction quality from the latent subspace of BeNeDiff is maintained given the behavioral labels. "Self-Supervised" denotes the VAE w/o behavior labels.

## 5.3 Neural Dynamics Exploration of Disentangled Latent Factors

From the quantitative experiments in the previous subsection, we obtained information about the decoding and disentanglement quality within the subspace. However, these metrics have limitations in interpreting single-trial neural dynamics, especially the complex temporal structures over time. Here, we visualize the generated videos from BeNeDiff and the baseline latent manipulation methods, demonstrating that BeNeDiff provides interpretable quantifications of the behaviors of interest.

**Latent Manipulation Methods for Comparison.** We compare the neural dynamics exploration performance of BeNeDiff against the following two latent manipulation methods:

• **Naïve Latent Manipulation**: the standard manipulation method discussed in Section 3.2.1, which approximates the posterior of behavioral videos given the neural latent trajectories $p(\mathbf{Y} \mid \mathbf{Z})$, using a neural network that incorporates recurrent units and spatio-temporal convolutional layers.

• **Classifier-free Guidance** [Ho and Salimans, 2022]: a method that approximates the posterior $p(\mathbf{Y} \mid \mathbf{Z})$ with diffusion models. It co-trains a conditional and an unconditional diffusion model together, combining the resulting conditional and unconditional scores at each diffusion step. In the conditional model, the entire neural latent trajectory $\mathbf{Z}$ is set as the condition, formulating the denoiser as $\hat{\boldsymbol{\epsilon}}(\mathbf{Y}_t, \mathbf{Z}, t)$. For the manipulation of the latent, we keep the activated latent factor $\mathbf{z}^{(d)}$ to evolve while setting the values of the other latent factors to those in the first frame of the trial.

**Setup.** To verify the neural dynamics interpretation capability of BeNeDiff, we generate behavioral video data given the activation of each behavior of interest (generated trials with the activation of Jaw and Paw-(y) are shown in Figure 6 and Figure 9, respectively). For visualization and video analysis, we plot frames at intervals of five and compute their frame differences. The conditional module of the classifier-free guidance method is trained with an auxiliary convolutional head. Compared to general video synthesis [Harvey et al., 2022; Esser et al., 2023], our behavioral video data are more focused on maintaining the temporal dynamics and consistency across video frames, thus in BeNeDiff, we tailor the standard 3D U-Net architecture [Çiçek et al., 2016] from temporal self-attention layers to temporal convolutions layers [Li et al., 2023b; 2024b] to maintain local temporal consistency. While we keep the spatial self-attention layers the same. The diffusion model is trained on an Nvidia V100, using approximately 20 computer hours.

**Results Analyses of the Generated Videos.** As shown in Figure 6, for the naïve latent manipulation method, the distribution of neural signals often falls outside the original distribution after manipulation, resulting in blurred generated frames. The frame differences are entangled, and the "Jaw" latent factor affects the entire head movement of the mouse, particularly in the first four frames shown.

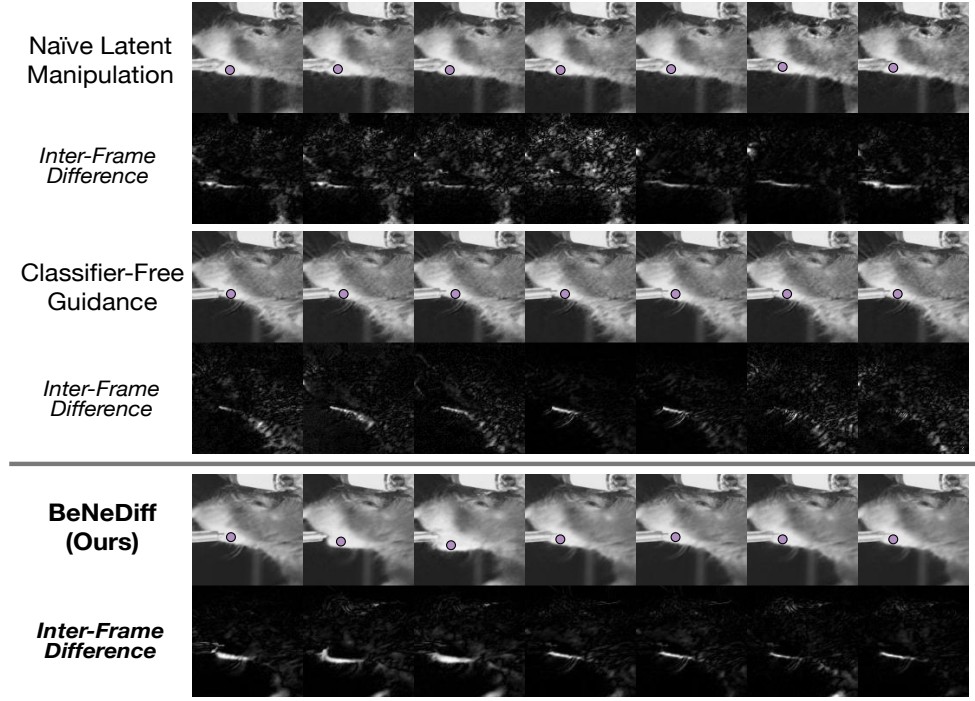

Figure 6: **Generated Single-trial Behavioral Videos with Latent Factor Guidance from the side view**. Compared to baseline methods, we observe that the neural dynamics of latent factor in the results of BeNeDiff show specificity to the "Jaw" movements.

On the other hand, for classifier-free guidance, the generated videos maintain coherent consistency between frames. However, it does not interpret neural dynamics well in this context, resulting in a trajectory with small movements in the "Jaw". This is because the overall latent trajectory is used as the input to the model and the other latent factors are kept fixed, making it difficult to discriminate the evolution of a single factor effectively. In contrast, the results of BeNeDiff show more specificity to the targeted behavior of interest. The inter-frame differences in BeNeDiff's results are clearly specified to the "Jaw" movements, and the structure of the neural dynamics is well-preserved and consistent with ground-truth "Jaw" behavior trajectories. A similar pattern is evident with the other latent factors, as shown in Figures 9, 10, and 11 in the appendix.

### 5.4 Neural Dynamics Exploration of Disentangled Latent Factors Across Brain Regions

Besides the capability of revealing interpretable neural dynamics of each latent factor associated with behaviors, here we further investigate the neural dynamics differences across brain regions through BeNeDiff. As shown in Figure 7 and Figure 12 in the appendix, we present the 2D neural latent trajectories of two latent factors, specifically related to "Paw-(x)" and "Paw-(y)", across six brain regions for two randomly selected trials. From the starting point of the trial, we observe that the latent trajectories corresponding to the left and right hemispheres of the VIS both show a noticeable change starting earlier. Next, the SSp regions show a large shift in activity, followed by a similar change in the MOs regions. However, it is difficult to clearly visualize the specific motion encoded by each region and to distinguish how different the motions are encoded solely based on neural trajectory plots. This further highlights the need for using a video diffusion model for visualization and interpretation.

In contrast, in the generated behavior video samples of BeNeDiff (as illustrated by the frame differences in Figure 8 and Figure 13 in the appendix), where the "Paw-(x)" and "Paw-(y)" latent factors are activated, the behavioral dynamics encoded by these two latent factors are observed across different brain regions. First, paw movements are detected in the VIS regions before the "Levers" come in. This early activity in VIS could reflect its role in the predictive coding of behaviors, indicating that this region may predict motor movements before they happen. Next, the SSp regions exhibit paw movements that are synchronized with the onset of the "Levers", indicating a potential role

for SSp in processing somatosensory feedback. Subsequently, in the MOs regions, paw movements are observed following the "Levers" onset, which is consistent with MOs' role in motor execution and control, occurring slightly after SSp.

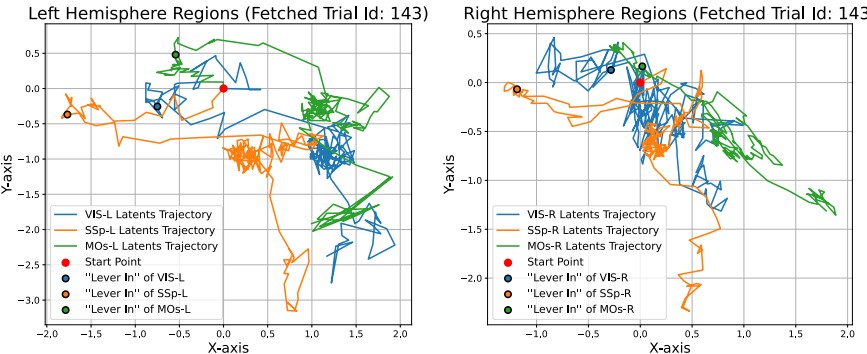

Figure 7: **Learnt Neural Latent Trajectories of BeNeDiff across various brain regions**. It is difficult to clearly visualize the specific motion encoded by each region and to distinguish how different the motions are encoded across brain regions.

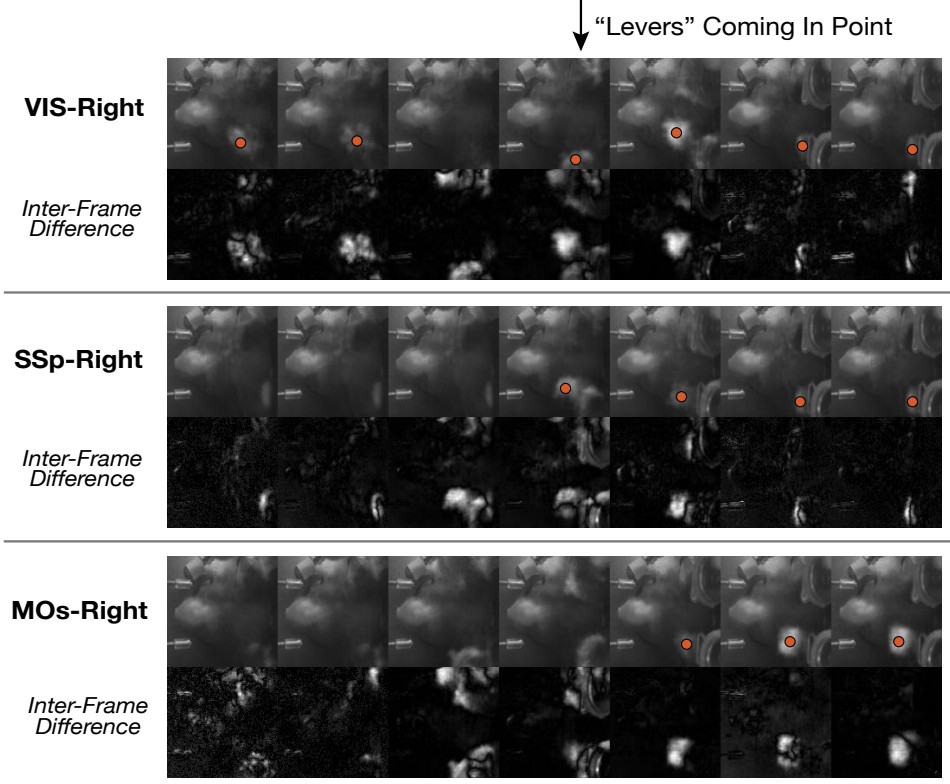

Figure 8: **Generated video frame differences across the right hemisphere regions**. The red dots in the figure indicate paw appearances.

To sum up, although neural trajectory plots provide a clear temporal sequence of activations across regions, it is challenging to directly visualize the specific behavioral dynamics encoded by each region and to discriminate how they differ. This limitation highlights the necessity for a video diffusion model in BeNeDiff, to better visualize and interpret the encoded behavioral dynamics of each neural latent factor. By synthesizing realistic behavior videos in a generative fashion, BeNeDiff enables us to better understand the unique neural dynamics in each brain region and their corresponding behavioral dynamics.

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

# Appendix to "Exploring Behavior-Relevant and Disentangled Neural Dynamics with Generative Diffusion Models"

## A    Methodology details

**Broader Impacts and Future Work.** Our results highlight the method's ability to reveal fine-grained neuro-behavioral relationships, advancing our understanding of how neural dynamics encode behavior. These results demonstrate how BeNeDiff can elucidate interpretable quantifications of behaviors of interest, making it a promising machine learning tool for explainable neuroscience. Future work will explore extending this approach to more neural datasets and further refining the generative models for more theoretical interpretability and utility in neuroscience research.

**Training Details of Neural LVM.** The neural signal dimensions for the brain regions are as follows: MOs_L: 14 dimensions, MOs_R: 14 dimensions, VIS_L: 24 dimensions, VIS_R: 21 dimensions, SSp_L: 23 dimensions, and SSp_R: 22 dimensions. Both the probabilistic encoder and decoder of the neural LVM are based on an RNN architecture Fabius and Van Amersfoort [2014]. Mean squared error (MSE) is used for both the neural reconstruction and behavior decoding loss. We use the Adam Optimizer Kingma and Ba [2014] for optimization and the learning rate is set as 0.001. The batch size is uniformly set to 32. The latent subspace factor number is fixed at 6, which is the same as the number of behaviors of interest. We employ the dropout technique Srivastava et al. [2014] and the ReLU activation function Rasamoelina et al. [2020] between layers in our probabilistic encoder and decoder neural networks.

**Training Details of Video Diffusion Models.** We adopt the architecture of the VDM of 3D-UNet [Ho et al., 2022b] with the $\epsilon$-parameterization. We use both spatial attention and spatial convolutions. The temporal convolutions are used to maintain consistency between frames. The embedding input size to the UNet architecture is set as 32 and the UNet has three downsampling and upsampling layers. The diffusion timestep is set as 200. The training batch size is set as 64, with a learning rate of 0.001. We use Group Normalization.

## B    In-depth Investigation on the neural LVM module across brain regions

Table 2: The $R^2\%$ and RMSE of the neural reconstruction, and the disentanglement MIG of the latent subspace on the VIS-Right region data. The boldface denotes the highest score of the MIG metric. Each experiment condition is repeated with 5 runs, and their mean and standard deviations are listed.

| Metrics \ Method | Session-1 | | Session-2 | |
|---|---|---|---|---|
| | Standard VAE | **Ours** | Standard VAE | **Ours** |
| $R^2(\%) \uparrow$ | 77.79 ($\pm$0.20) | 73.74 ($\pm$0.24) | 78.68 ($\pm$0.21) | 71.13 ($\pm$0.29) |
| RMSE $\downarrow$ | 48.94 ($\pm$0.18) | 55.17 ($\pm$0.19) | 49.54 ($\pm$0.22) | 54.27 ($\pm$0.19) |
| MIG($\%$) $\uparrow$ | 34.61 ($\pm$0.30) | **56.36** ($\pm$0.29) | 33.20 ($\pm$0.27) | **59.05** ($\pm$0.26) |

Table 3: **Ablation Study** of the neural LVM module. The boldface denotes the highest score of the MIG metric. Each experiment condition is repeated with 5 runs, and their mean and standard deviations are listed.

| Region | Metrics | Standard VAE | w/o Beha | w/o TC | **Ours** |
|---|---|---|---|---|---|
| VIS-Left | $R^2(\%) \uparrow$ | 83.66 ($\pm$0.21) | 77.74 ($\pm$0.23) | 79.82 ($\pm$0.29) | 75.41 ($\pm$0.24) |
| | RMSE $\downarrow$ | 30.96 ($\pm$0.18) | 34.86 ($\pm$0.20) | 34.71 ($\pm$0.13) | 35.50 ($\pm$0.17) |
| | MIG($\%$) $\uparrow$ | 33.13 ($\pm$0.24) | 48.54 ($\pm$0.23) | 38.13 ($\pm$0.27) | **55.87** ($\pm$0.26) |
| MOs-Left | $R^2(\%) \uparrow$ | 84.70 ($\pm$0.24) | 76.08 ($\pm$0.20) | 75.49 ($\pm$0.22) | 69.59 ($\pm$0.22) |
| | RMSE $\downarrow$ | 31.41 ($\pm$0.22) | 34.14 ($\pm$0.25) | 34.92 ($\pm$0.16) | 36.91 ($\pm$0.18) |
| | MIG($\%$) $\uparrow$ | 32.96 ($\pm$0.21) | 49.79 ($\pm$0.23) | 40.74 ($\pm$0.23) | **58.56** ($\pm$0.29) |

# C  Video Generation Results on Various Behaviors of Interests

Using Figure 9 as an example, for the naïve latent manipulation method, the generated frames are in a reasonable form. Nevertheless, the frame differences are still intertwined, and the latent factor of "Paw-(y)" heavily affects the "Spout" movement. Meanwhile, for classifier-free guidance, the trajectories focus on the mouse movements, but they are still entangled with the "Chest" movements. In contrast, the results of BeNeDiff show more specificity to the targeted behavior of interest. The inter-frame differences in BeNeDiff's results are clearly specified to the "Paw-(y)" movements, and the temporal evolution of the neural dynamics is coherent with real-world mouse paw trajectories. The generated results in Figures 10 and 11 show a similar trend, demonstrating specificity to the Paw-(x)" and Spout" factors.

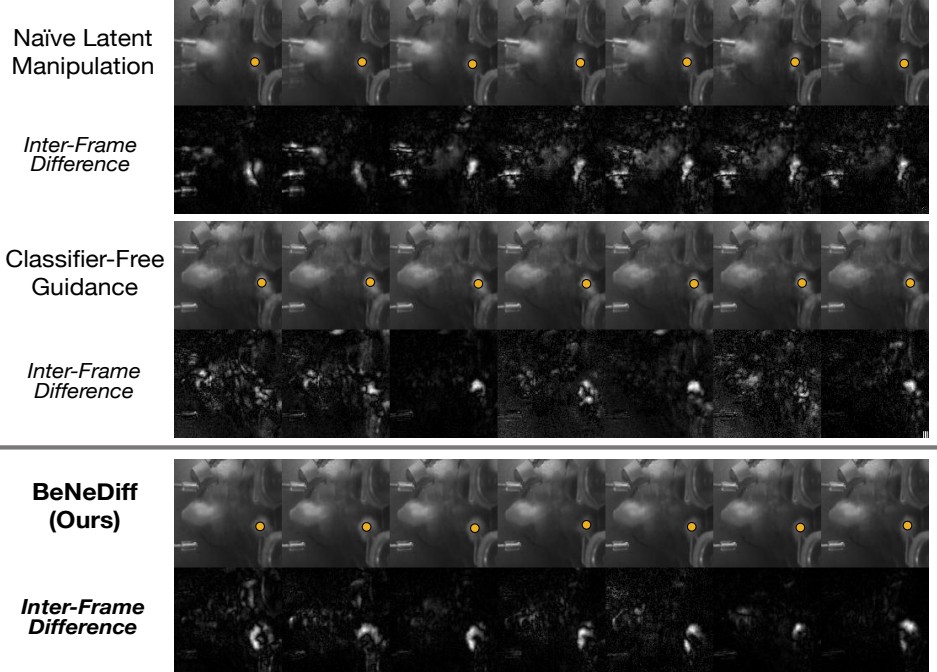

Figure 9: **Generated Single-trial Behavioral Videos with Latent Factor Guidance from the bottom view**. Compared to baseline methods, we observe that the neural dynamics of a latent factor in the results of BeNeDiff show specificity to the "Paw-(y)" movements.

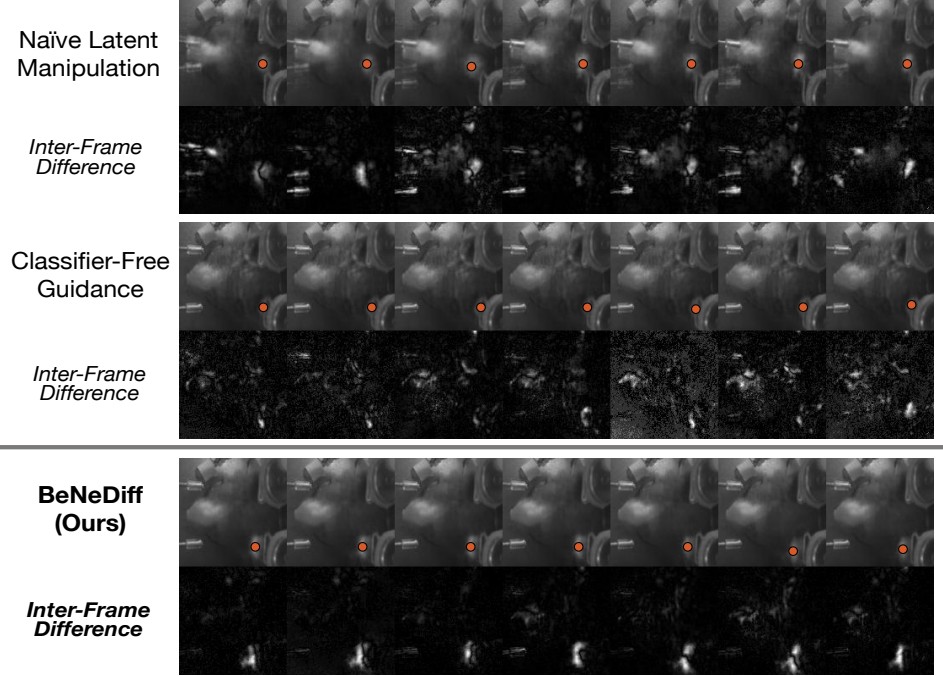

Figure 10: **Generated Single-trial Behavioral Videos with Latent Factor Guidance from the bottom view**. Compared to baseline methods, we observe that the neural dynamics of a latent factor in the results of BeNeDiff show specificity to the "Paw-(x)" movements.

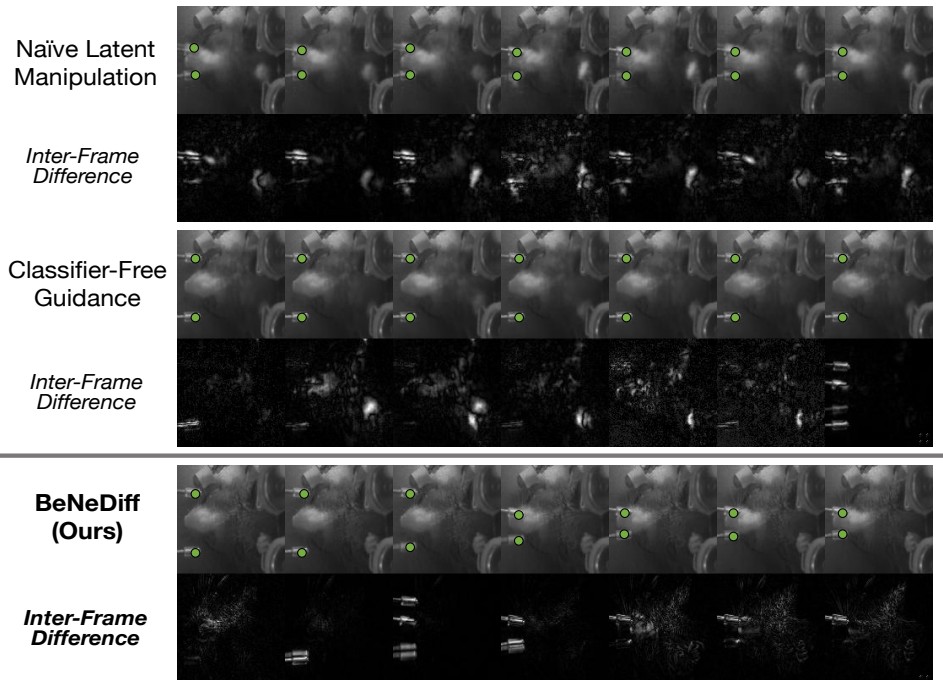

Figure 11: **Generated Single-trial Behavioral Videos with Latent Factor Guidance from the bottom view**. Compared to baseline methods, we observe that the neural dynamics of a latent factor in the results of BeNeDiff show specificity to the "Spout" movements.

## D   Learnt Neural Latent Trajectories of BeNeDiff across various brain regions

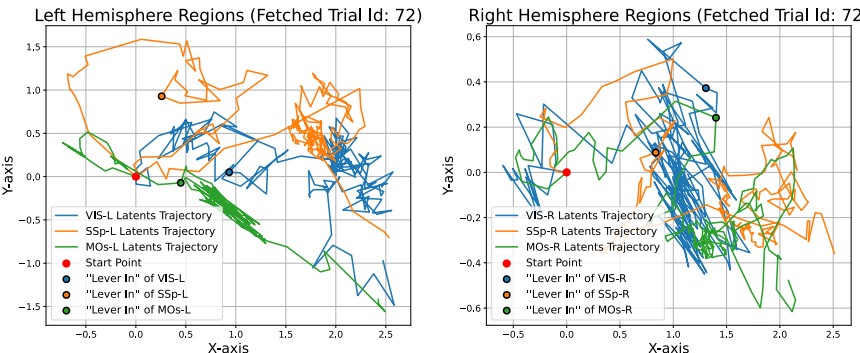

Figure 12: **Learnt Neural Latent Trajectories of BeNeDiff across various brain regions**. It is difficult to clearly visualize the specific motion encoded by each region and to distinguish how different the motions are encoded across brain regions.

## E   Video Generation Results on Various Brain Regions of the Left Hemisphere

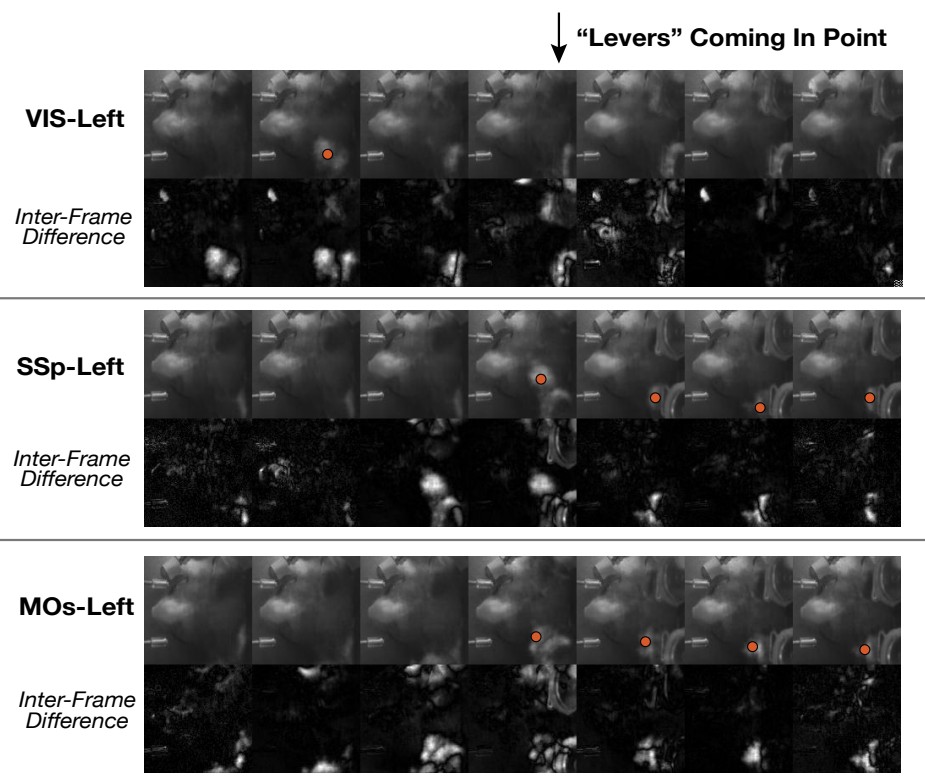

Figure 13: **Generated video frame differences across the left hemisphere regions**. The red dots in the figure indicate paw appearances.

# F   Discussion and Limitation

Our study introduces BeNeDiff, a novel approach leveraging behavior-informed latent variable models and generative diffusion models to uncover and interpret neural dynamics. Through empirical validation, we demonstrate that BeNeDiff effectively identifies a disentangled neural subspace and synthesizes behavior videos that provide interpretable insights into neural activities associated with distinct behaviors of interest. However, for the neural latent variable model (LVM) module, there exists a balance between disentangling the neural subspace with behavior semantics and maintaining neural reconstruction performance. For each brain region and session, at this stage, a careful hyper-parameter search is necessary to balance the weight between these two components. For the generative video diffusion module, we implement the neural encoder (classifier for guidance) as a linear regressor for interpretability. This linear assumption can be relaxed later for improved guidance performance.

