# OpenReview forum: "Exploring Behavior-Relevant and Disentangled Neural Dynamics with Generative Diffusion Models"
_NeurIPS.cc/2024/Conference — NeurIPS 2024 poster_

### Official Review · Reviewer_HCYS · 2024-06-15

**Soundness:** 3
**Presentation:** 4
**Contribution:** 3
**Rating:** 7
**Confidence:** 4

**Summary:**

This study presents BeNeDiff that integrates behavior-informed latent variable models with state-of-the-art generative diffusion models to investigate neural dynamics during behavioral tasks. The methodology involves identifying fine-grained and disentangled neural subspaces and synthesizing behavior videos to provide interpretable quantifications of neural activity related to specific behaviors. The authors validate their approach using multi-session widefield calcium imaging datasets, demonstrating high levels of neural disentanglement and reconstruction quality.

**Strengths:**

1. Using wide-field calcium imaging for neural-behavior analysis is both innovative and interesting.

2. The approach of employing a behavior LVM for latent factor disentanglement, followed by a video diffusion model for behavior encoding, is both clear and impressive.

3. Especially, using $Z$ produced by LVM as learning target for the classifier in video diffusion model is quiet important, avoiding the issue of "label leakage" which maybe introduced by classifier-free guidance generation.

4. The paper is well-organized, and the disentanglement results and behavior encoding results are both meaningful and useful.

**Weaknesses:**

1. This work primarily relies on a behavior LVM for neural activity disentanglement, which can be influenced by the newly introduced total correlation (TC) penalty term with $\beta$ weighted. The impact of this penalty should be considered through ablation studies.


**Not a Weakness**: There is significant research combining behavioral labels with latent variables of neural activity [1,2], considering the effects of different disentanglement algorithms on behavior encoding would further enhance this study.

[1] Zhou, Ding, and Xue-Xin Wei. "Learning identifiable and interpretable latent models of high-dimensional neural activity using pi-VAE." Advances in Neural Information Processing Systems 33 (2020): 7234-7247.

[2] Schneider, Steffen, Jin Hwa Lee, and Mackenzie Weygandt Mathis. "Learnable latent embeddings for joint behavioural and neural analysis." Nature 617.7960 (2023): 360-368.

**Questions:**

I have no question.

**Limitations:**

see WeaKnesses part.

---

> ### Author Rebuttal · Authors · 2024-08-07
>
> Dear Reviewer HCYS,
>
> Thank you for your recognition of the work and insightful questions. We provide clarifications and new results that we have generated to address your questions below.
>
> > This work primarily relies on a behavior LVM for neural activity disentanglement, which can be influenced by the newly introduced total correlation (TC) penalty term with 𝛽 weighted. The impact of this penalty should be considered through ablation studies.
>
> We appreciate your attention to this aspect. The following are our ablation studies with various $\beta$ values. Each experiment condition is repeated with 5 runs, and their mean and variance are listed.
>
> |              Metrics               |    $\beta = 1$     |    $\beta = 2$     | $\beta = 4$ (Ours) |    $\beta = 8$     |
> | :--------------------------------: | :----------------: | :----------------: | :----------------: | :----------------: |
> |        $R^2$(%) $\uparrow$         | $79.24 (\pm 0.27)$ | $77.97 (\pm 0.19)$ | $75.41 (\pm 0.24)$ | $71.44 (\pm 0.28)$ |
> |      RMSE (x100) $\downarrow$      | $33.90 (\pm 0.21)$ | $34.65 (\pm 0.16)$ | $35.50 (\pm 0.17)$ | $38.84 (\pm 0.18)$ |
> | $\operatorname{MIG}$(%) $\uparrow$ | $48.75 (\pm 0.26)$ | $51.37 (\pm 0.28)$ | $55.87 (\pm 0.26)$ | $58.85 (\pm 0.26)$ |
>
> *Table 1: Hyper-parameter $\beta$ Investigation on the SSp-Left region of Session-1.*
>
> |              Metrics               |    $\beta = 1$     |    $\beta = 2$     | $\beta = 4$ (Ours) |    $\beta = 8$     |
> | :--------------------------------: | :----------------: | :----------------: | :----------------: | :----------------: |
> |        $R^2$(%) $\uparrow$         | $74.93 (\pm 0.24)$ | $72.50 (\pm 0.20)$ | $69.59 (\pm 0.22)$ | $66.00 (\pm 0.27)$ |
> |      RMSE (x100) $\downarrow$      | $33.80 (\pm 0.16)$ | $35.08 (\pm 0.22)$ | $36.91 (\pm 0.18)$ | $39.99 (\pm 0.18)$ |
> | $\operatorname{MIG}$(%) $\uparrow$ | $49.26 (\pm 0.28)$ | $52.76 (\pm 0.29)$ | $58.56 (\pm 0.29)$ | $61.06 (\pm 0.27)$ |
>
> *Table 2: Hyper-parameter $\beta$ Investigation on the MOs-Left region of Session-1.*
>
> From the results, we observed that setting $\beta = 1$ or $\beta = 2$ resulted in lower disentanglement of the latent subspace. However, increasing $\beta = 8$ led to substantial degradation in neural reconstruction quality. Hence, we determined that $\beta = 4$ offers a balanced trade-off between disentanglement and reconstruction. These Tables will be added in the Appendix of the revised manuscript.
>
> > There is significant research combining behavioral labels with latent variables of neural activity [1,2], considering the effects of different disentanglement algorithms on behavior encoding would further enhance this study.
>
> This is an insightful point. We would like to note that the listed two research works [1, 2] combining behavioral labels for neural latent variable learning do not focus on the disentanglement of the neural subspace. The total-correlation (TC) term proposed here is promising to be combined with these two approaches [1, 2] to achieve a more disentangled latent subspace.  We added the TC term to their loss functions, and the results are reported in the following Tables 3 and 4. Each experiment condition is repeated with 5 runs, and their mean and variance are listed.
>
> |          Metrics \ Method          |       pi-VAE       |    pi-VAE w/ TC    |       CEBRA        |    CEBRA w/ TC     |
> | :--------------------------------: | :----------------: | :----------------: | :----------------: | :----------------: |
> |        $R^2$(%) $\uparrow$         | $79.60 (\pm 0.22)$ | $72.10 (\pm 0.25)$ | $74.37 (\pm 0.24)$ | $70.51 (\pm 0.28)$ |
> |      RMSE (x100) $\downarrow$      | $33.07 (\pm 0.18)$ | $36.20 (\pm 0.20)$ | $36.74 (\pm 0.22)$ | $37.82 (\pm 0.24)$ |
> | $\operatorname{MIG}$(%) $\uparrow$ | $40.12 (\pm 0.24)$ | $57.27 (\pm 0.23)$ | $43.98 (\pm 0.29)$ | $49.66 (\pm 0.27)$ |
>
> *Table 3: Total-correlation term effect analyses on the SSp-Left region of Session-1.*
>
> |          Metrics \ Method          |       pi-VAE       |    pi-VAE w/ TC    |       CEBRA        |    CEBRA w/ TC     |
> | :--------------------------------: | :----------------: | :----------------: | :----------------: | :----------------: |
> |        $R^2$(%) $\uparrow$         | $72.63 (\pm 0.28)$ | $66.99 (\pm 0.29)$ | $70.73 (\pm 0.23)$ | $67.65 (\pm 0.23)$ |
> |      RMSE (x100) $\downarrow$      | $32.14 (\pm 0.17)$ | $37.12 (\pm 0.18)$ | $35.69 (\pm 0.19)$ | $37.75 (\pm 0.25)$ |
> | $\operatorname{MIG}$(%) $\uparrow$ | $37.94 (\pm 0.23)$ | $60.14 (\pm 0.24)$ | $42.20 (\pm 0.28)$ | $50.21 (\pm 0.31)$ |
>
> *Table 4: Total-correlation term effect analyses on the MOs-Left region of Session-1.*
>
> After incorporating the total-correlation term,  we observe that the disentanglement performance of these baseline neural LVMs improves. These results will also be added to the Appendix of the revised manuscript. We would like to emphasize that the disentangled neural LVM module is not our primary contribution. Instead, the main modeling and scientific contribution of our work both lies in the video diffusion modeling (VDM) module of BeNeDiff, which interprets the neural dynamics of each disentangled latent factor in a generative manner. The VDM module is crucial for clearly interpreting the neural dynamics of each latent factor, demonstrating specificity to behaviors of interest (e.g., paw-x-axis movement, jaw movement) and aligning with ground-truth behavioral trajectories.  Moreover, the VDM module can be generalized to all the aforementioned neural LVM baselines with the total-correlation term for the discovery of neural dynamics in the learned neural subspace. We can also include more generated video results in an anonymized link upon request.
>
>
>
> Refs:
>
> [1] Learning identifiable and interpretable latent models of high-dimensional neural activity using pi-VAE. (Zhou et al., 2020)
>
> [2] Learnable latent embeddings for joint behavioural and neural analysis. (Schneider et al., 2023)

---

> > ### Comment · Reviewer_HCYS · 2024-08-09
> >
> > Thank you for the author's response. The new information meets my expectations, and I hope this content can be added to the supplementary materials. Thus, I maintain my original score.

---

> > > ### Author Response · Authors · 2024-08-09
> > > **Thank you**
> > >
> > > Dear Reviewer HCYS,
> > >
> > > Thank you for your timely response. We will include the ablation study of the hyperparameter $\beta$ within the total-correlation term and present the results in the appendix of the revised manuscript. Additionally, we will assess the generalizability of the video diffusion modeling (VDM) module to the pi-VAE and CEBRA neural LVM baselines (with the total-correlation penalty term) to discover neural dynamics in their learned disentangled subspace. The generated video results (similar to Figures 6 and 7 of the manuscript) will be  added to the supplementary materials.
> > >
> > > We again appreciate your evaluation and recognition of our work.

---

### Official Review · Reviewer_CLKJ · 2024-07-12

**Soundness:** 3
**Presentation:** 2
**Contribution:** 3
**Rating:** 6
**Confidence:** 3

**Summary:**

The paper proposed BeNeDiff to learn disentangled neural trajectories together with a generative diffusion mode for behavior. BeNeDiff leverages beta-VAE for learning of disentangled space space Additional behavior generation module is applied for interpretation of the neural subspace. It is interesting to see that the learned latent latent trajectory is separated for different behavior types. The qualitative results and quantitative results demonstrate the superior of the proposed method.

**Strengths:**

1.The disentangle property of neural latent space is significant compared to previous methods based on the quantitative metric.

2.The idea leveraging diffusion model to interpret learned neural subspace is interesting.

**Weaknesses:**

1.It is unclear to me if the behavior video generation part is trained together with the disentangled subspace learning. It seems true based on the figure, but didn't see how they are optimized together from text.

2.it is interesting to learn the disentangled neural space. Some visualization of the learned latent space is beneficial.

3.Additional ablation study on different contribution of the loss function could be performed.

**Questions:**

1.How training and testing data are split? Are they session wise or random? How the method can be generalized to different sessions or animals?

2.Does the learned latent trajectory show some clustering property, or somehow organized, could you visualize it?

3.For session 3.2, $c$ is better to be described as it is first mentioned in eq.5. Does the number of class is equals to feature dimension D and the same as the number of the behavior classes"?

4.To get figure 4, which latent variable do you use, $\hat{Z}$ or $Z$?

5.For fig.6, and fig.7, could you show the difference of the ground truth frame with the same behavior type as comparison.

**Limitations:**

It could be better the paper include more analyze on the learned latent trajectory and see if it could provide some scientific insight.

---

> ### Author Rebuttal · Authors · 2024-08-07
>
> Dear Reviewer CLKJ,
>
> Thank you for your valuable comments. We would like to make the following clarifications. Hopefully these will resolve most of your concerns, and they can be taken into account when deciding the final review score.
>
> > It is unclear to me if the behavior video generation part is trained together with the disentangled subspace learning. It seems true based on the figure, but didn't see how they are optimized together from text.
>
> We would like to note that given the learned disentangled neural subspace and trajectories (from the neural latent variable model), the behavioral video (generation) diffusion model (VDM) module serves as an expressive ML tool for interpreting and visualizing complex factor-wise neural dynamics. Hence, the behavioral VDM is trained after the neural LVM in two steps, as we need high-quality neural latents for training the classifier (neural encoder) in VDM. This procedure is also in line with the conventions of previous studies [1] for scientific interpretation.
>
> Regarding Figure 2, it describes the neural dynamics interpretation phase of BeNeDiff rather than the model training phase. In this phase, behavior videos are generated with guidance from the neural encoder. In the revised manuscript, we will also enhance Figure 2 for clearer visualization.
>
> > It is interesting to learn the disentangled neural space. Some visualization of the learned latent space is beneficial. Does the learned latent trajectory show some clustering property, or somehow organized, could you visualize it? It could be better the paper include more analyze on the learned latent trajectory and see if it could provide some scientific insight.
>
> We thank the reviewer for this practical suggestion. In Figure 1(B) of the attached PDF, in each subfigure, we select two neural latent factors at a time and plot their 2D trajectories. Latent-factor-3 is informed by the "Paw-x" behavior-of-interest. We observe that the learned dimension axes are disentangled, but the behavior dynamics are not readable from the trajectories. Figure 1(C) depicts the temporal evolution of all six latent factors, which is also difficult to interpret in terms of the behavior dynamics encoded by the neural population. In Figure 1(D) of the attached PDF, the full neural latent trajectories of trials (dimension reduction by PCA to 3D) are also hard to directly interpret or identify any clustering structure.
>
> To sum up, the learned neural subspace exhibits certain disentangled properties, but the visualization results are not intuitive or readable enough for experimentalists. Therefore, the VDM module in BeNeDiff is crucial for clearly interpreting the neural dynamics of each latent factor, which demonstrates specificity to behaviors of interest and aligns with ground-truth behavioral trajectories (e.g., Latent-factor-3 is aligned with "Paw-x" in Figure 1(E)). Please refer to Section 1.2 of the global response for detail.
>
> > Additional ablation study on different contribution of the loss function could be performed.
>
> Please refer to Section 1.4 of the global response for a detailed discussion of this point.
>
> > (1) How training and testing data are split? Are they session wise or random? (2) How the method can be generalized to different sessions or animals?
>
> (1) We train separate models for each session independently. Within each session, we use an 80/20 split for training and testing, employing cross-validation to ensure model robustness.
>
> (2) This is an interesting point. The generalizability of each method is a vital consideration. For the neural data part, the difficulty arises from the varying number of observed neurons across different sessions. Additionally, the same brain region can exhibit varying encoding for the same behavior of interest across different animals. One possible solution to mitigate the issues is to add a linear probing layer to perform neural align [2, 3] from various sessions and animals into a unified subspace, then can be adpated to the BeNeDiff framework. Meanwhile, the VDM module can generalize well to multi-sessions and multi-animals settings by fitting a unified neural encoder for the aligned neural data.
>
> > (1) For session 3.2, 𝑐 is better to be described as it is first mentioned in eq.5. (2) Does the number of class is equals to feature dimension D and the same as the number of the behavior classes"?
>
> (1) Thank you for this practical suggestion. We will move the description of class label $\mathbf{c}$ from line 147 to after Eq. (5).
>
> (2) Yes. The number of class labels $\mathbf{c}$ is the same as the latent factor dimension $D$, as well as the number of behaviors of interest.
>
> >  To get figure 4, which latent variable do you use, 𝑍^ or 𝑍?
>
> We use each latent factor of $\mathbf{Z}$ for behavior decoding, and get the results in Figure 4.
>
> >  For fig.6, and fig.7, could you show the difference of the ground truth frame with the same behavior type as comparison.
>
> We apologize for the confusion and would like to clarify that the results in Figures 6 and 7 are all derived from analyzing the behavior-related neural dynamics of one target neural latent factor using two baseline latent manipulation methods and BeNeDiff. The ground truth frames from behavior trials in the dataset do not specifically activate one neural latent factor, so the frame differences between consecutive images result in entangled behaviors. Including them would not provide a fair comparison. We have, however, provided several ground-truth behavior trials in the supplementary materials for reference.
>
>
>
> We look forward to further discussion, and are happy to answer any questions that may arise.
>
>
>
> Refs:
>
> [1] Cortical discovery using large scale generative models. (Luo et al., 2023)
>
> [2] Stabilizing brain-computer interfaces through alignment of latent dynamics. (Karpowicz et al., 2022)
>
> [3] Using adversarial networks to extend brain computer interface decoding accuracy over time. (Ma et al., 2023)

---

> > ### Comment · Reviewer_CLKJ · 2024-08-10
> >
> > Thanks for the responses. The responses addressed most of my concerns.
> >
> > But towards Q2, if the number of latent factor $D$ and the number of behavior of interest are the same, use the class label as guidance would obviously be helpful to learn a disentangled latent space. Directly learning a classifier and use the latent from the classifier would also be disentangled.
> >
> > It would be good to provide the latent variables from the trained classifier to diffusion model to see if the behavior videos could be synthesized from these variables.

---

> > > ### Author Response · Authors · 2024-08-12
> > > **Thank you**
> > >
> > > Dear Reviewer CLKJ,
> > >
> > > We appreciate your positive feedback and the adjustment to your score. As mentioned in our previous response, we will include the video results synthesized from behavior classifier latents in the supplementary materials of the revised manuscript. Thank you once again for your valuable suggestions and comments.

---

> ### Author Response · Authors · 2024-08-12
> **Synthesizing Videos with Behavior Classifier Latents**
>
> Dear Reviewer CLKJ,
>
> We thank you for the valuable response. We agree that incorporating behavior labels can be helpful for learning a disentangled neural subspace. However, we note that our neural LVM module in BeNeDiff maintains a high neural data reconstruction rate (as shown in Table 1 of the manuscript), thereby preserving the neural dynamics in the latent trajectories.
>
> It is true that we can train a probabilistic behavior classifier (a behavior decoder from neural data $\mathbf{X}$ to behavior labels $\mathbf{U}$) to approximate the conditional distribution $p(\mathbf{U} \mid \mathbf{X})$ and derive latent variables $\mathbf{Z}'$ from it. The $\mathbf{Z}'$ would also be disentangled. However, such $\mathbf{Z}'$ lacks physical or semantic meaning, as there is no reconstruction of the neural data, making it unable to interpret neural dynamics or provide any scientific insights. We further trained a diffusion model guided by $\mathbf{Z}'$ and presented the visualization results. We have sent the AC a comment linking to an anonymous repository containing these synthesized video results. In accordance with this year’s NeurIPS guidelines, please have the AC forward it to you.
>
> We observe that while the synthesized videos by activating each latent factor does show some specificity to a particular behavior of interest, **the overall video results are quite chaotic and do not align with the ground-truth behavioral trajectories.** **The behavior movements are overly drastic and contain many unnatural distortions** compared to the videos generated by the neural latent trajectories in BeNeDiff. These videos will be included in the supplementary materials of the revised manuscript. Additionally, we have included the videos generated by BeNeDiff and the ground-truth behavior videos in the anonymous repository provided to the AC for comparison.
>
> Looking ahead, our future research aims to generalize BeNeDiff to settings where the latent (feature) dimension is larger than the number of behavior classes. We hope that these responses and the above rebuttals address your concerns and will be considered when determining the final review score.

---

### Official Review · Reviewer_o3cJ · 2024-07-15

**Soundness:** 2
**Presentation:** 3
**Contribution:** 3
**Rating:** 5
**Confidence:** 3

**Summary:**

The authors implement a VAE with behavioral labels to identify behavior-specific latent variables. They use these latents to then create a generative model for experimental video. They focus on the application of their model to a wide-field cortical recordings of a mouse during an experimental visual task. They show that their model identifies disentangled behaviorally relevant latents and can create experimental video where identifiable behavioral elements are generated from specific neural latents.

**Strengths:**

I see two primary contributions in this paper that make it appropriate for a Neurips audience. 1) The use of Total correlation measures to disentangle latents in a VAE framework in neuroscience. 2) Using generative video diffusion models to identify what video features relate to individual neural latents. Both of these are interesting applications of existing methods but they are applied in a clever way to neural data and I think these would be of interest to a computational neuroscience audience.

**Weaknesses:**

Each of the primary contributions to neural LVM type models (the use of total correlation and the diffusion modeling) are not sufficiently benchmarked against competing approaches. It is unclear how this model compares to the others used in neuroscience. If the authors wanted to strengthen the case that their model is an important novel contribution for behaviorally-relevant latent disentangling, they could compare how total correlation VAE compares to other methods where behavioral labels are used to help disentangle the latent states. One important comparison that I would be curious to see that isn't cited would be the pi-VAE1, but the others the authors cite could be used. Specifically, these other models could be added to figures 4 and 5.

The authors could also focus on how using video diffusion is a novel application in this setting. Again, some additional benchmarking with existing deep LVMs in neuroscience and some further discussion and emphasis as to how the diffusion methodology solves an important problem in neuroscience that perhaps existing methods cannot solve would strengthen this paper.

Though the authors highlight the application of the model to this specific dataset as a contribution, this alone would be more appropriate for a more experimentally-focused venue. For neurips, the focus should be on the model advancements and their applications, not the significance of the dataset.

1 - Learning identifiable and interpretable latent models of high-dimensional neural activity using pi-VAE - Zhou and Wei

**Questions:**

NA

**Limitations:**

The authors only minimally discussed the limits of the ability of their model.

---

> ### Author Rebuttal · Authors · 2024-08-07
>
> Dear Reviewer o3cJ,
>
> Thank you for your detailed and constructive comments. We would like to make the following clarifications. Hopefully these will resolve most of your concerns, and they can be taken into account when deciding the final review score.
>
> > Each of the primary contributions to neural LVM type models (the use of total correlation and the diffusion modeling) are not sufficiently benchmarked against competing approaches.
>
> We would like to clarify that the contribution of BeNeDiff is two-fold: (1) The use of the total correlation term in the behavior-informed neural latent variable model (LVM) to induce a disentangled neural latent subspace. While this module is not our main modeling focus, it is effective compared to baselines (please refer to Table 1 of the attached PDF for the results). (2) The main novelty and contribution of our paper is the video diffusion modeling (VDM) module, which serves as an interpretation tool for visualizing the neural dynamics of each learned disentangled latent factor. The generated behavioral videos of VDM conditioned on activating one learned latent factor over time demonstrate that each factor shows specificity to a ground-truth behavior of interest (e.g., paw-x-axis movement, jaw movement) and aligns with ground-truth behavioral trajectories. The qualitative performance comparison of this VDM module against baselines is presented in Figures 6 and 7 of the manuscript, with detailed analyses in Sections 3.2.1 and 5.3.
>
> > If the authors wanted to strengthen the case that their model is an important novel contribution for behaviorally-relevant latent disentangling, they could compare how total correlation VAE compares to other methods where behavioral labels are used to help disentangle the latents. One important comparison that I would be curious to see would be the pi-VAE [1].
>
> Including the suggested pi-VAE [1], we have added the following three baselines for extensive comparisons. Please refer to Section 1.3 of the global response for the baseline descriptions and tables with results. We also note that the VDM module of BeNeDiff can generalize to interpret the neural dynamics of the latent factors learned by all the neural LVM baselines mentioned above.
>
> > The authors could also focus on how using video diffusion is a novel application in this setting. Again, some further discussion and emphasis as to how the diffusion methodology solves an important problem in neuroscience that perhaps existing methods cannot solve would strengthen this paper.
>
> This is a highly valid point. We would like to emphasize that the *scientific question* in neuroscience we aim to answer in this paper is "*how can we enable in-depth exploration of neural latent factors with video behavior recorded, revealing interpretable neural dynamics associated with behaviors*". Therefore, the modeling goal is to develop a machine learning tool for visualizing neural dynamics encoded by each learned latent factor. This involves mapping the neural latent factors $\mathbf{Z}$ to behavior videos $\mathbf{Y}$, activating a single latent factor $\mathbf{z}^{(d)}$, and observing how the manipulation affects $\mathbf{Y}$. The induced changes in the videos reveal the dynamics encoded by $\mathbf{z}^{(d)}$.
>
> Previous manipulation methods [2, 3] often change the target $\mathbf{z}^{(d)}$ while fixing non-target subspaces to arbitrary values. However, setting arbitrary values without knowing the true distributions of non-target subspaces can lead to unnatural distortions in generated videos, complicating the interpretation and visualization of genuine animal behavioral dynamics.
>
> BeNeDiff proposes using a video diffusion model (VDM) module to explore disentangled neural dynamics in a generative manner. Specifically, the VDM module employs a neural encoder (Eq. (7) of the manuscript) to guide the generation of videos that maximize variance in the neural trajectory of target latent factor $\mathbf{z}^{(d)}$ while minimizing variance in other factors' trajectories. This approach ensures that the generated behavioral videos predominantly reflect the neural dynamics of the target latent factor. It maintains naturalness by avoiding assumptions about specific values in non-target subspaces, thereby preventing the generation of videos with unnatural distortion, which manifests the modeling-wise novelty of BeNeDiff. As shown in Figures 6 and 7 of the manuscript, for each latent factor, the VDM module provides interpretable quantifications of its neural dynamics with specificity to the behaviors of interest. These generated video results are also available in the 'BeNeDiff-Generated-Video' folder of the supplementary materials.
>
> > Though the authors highlight the application of the model to this specific dataset as a contribution, this alone would be more appropriate for a more experimentally-focused venue. For neurips, the focus should be on the model advancements and their applications.
>
> We would like to politely disagree with the statement that the application of our proposed method is specific to this dataset. Please refer to Section 1.1 of the global response for details on how BeNeDiff can be generalized to multiple datasets. For the model advancements, please refer to the answer of the above point. We also note that the diffusion model has been employed as an interpretation tool in previous neuroscience works, resulting in publication [4] at NeurIPS.
>
> > The authors only minimally discussed the limits of the ability of their model.
>
> Due to space limitations, please refer to Section 1.5 of the global response for detail.
>
>
>
> We look forward to further discussion, and are happy to answer any questions that may arise.
>
>
>
> Refs:
>
> [1] pi-VAE. (Zhou et al., 2020)
>
> [2] Partitioning variability in animal behavioral videos using ss-vaes. (Whiteway et al., 2021)
>
> [3] Classifier-free diffusion guidance. (Ho et al., 2022)
>
> [4] Cortical discovery using large scale generative models. (Luo et al., 2023)

---

> > ### Author Response · Authors · 2024-08-10
> > **Thank you**
> >
> > Dear Reviewer o3cJ,
> >
> > Thank you for your positive feedback and for adjusting your score. As mentioned in our rebuttal, we will incorporate the neural LVM module comparisons and the VDM module-generated video visualizations for multiple datasets into the appendix of the revised manuscript.

---

> > ### Comment · Reviewer_o3cJ · 2024-08-10
> >
> > Thank you for the response. I think with this added detail the paper's contribution is clearer and I'd suggest emphasizing and clarifying the contribution similarly in a revised manuscript. I also appreciate the additional benchmarking with the total correlation measure. I have adjusted my score accordingly.
> >
> > Regarding my comment about your emphasis on the data set - I did not mean to suggest that the analysis is specific to this dataset, but rather that your phrasing in the manuscript emphasizes the data set in-and-of-itself as a major contribution. The end of the introduction reads "To highlight our major contributions: (1) This is the first work to explore wide-field imaging of
> > the dorsal cortex of mice during a decision-making task using neural subspace analysis," and on like 167 "However, our work is the first to discover interpretable and disentangled latent subspaces of wide-field imaging data". I agree that your methods are generally applicable across datasets and I believe, therefore, that the model and it's generality should be what is primarily emphasized when discussing the major contributions in the manuscript.

---

> ### Author Response · Authors · 2024-08-12
> **Thank you**
>
> Dear Reviewer o3cJ,
>
> We sincerely appreciate your positive evaluation of our method's contribution and additional benchmarking. We apologize for the confusion in the Introduction section and will clarify the model's generalizability across datasets when discussing the major contributions in the revised manuscript. We thank you once again for the valuable feedback and suggestions.

---

### Author Rebuttal · Authors · 2024-08-07

We would like to express our sincere gratitude to all the reviewers for their insightful feedback and suggestions. We appreciate the positive comments which characterized our work as having an `"interesting and clever"` idea for leveraging video diffusion models to interpret inferred neural subspaces  (o3cJ, CLKJ), `"significant and impressive"` methodological advancement (CLKJ, HCYS), being `"appropriate for NeurIPS and neuroscience audiences"` (o3cJ), with a `"well-organized presentation"` (tfQ5, 2uTS), and recognizing the experimental results of our work to be `"meaningful and useful"` (HCYS).

In this place, we would like to first provide several general clarifications to enhance overall understanding of our work.

**1.1 Generalizability of BeNeDiff to Datasets**

We would like to clarify that our framework, BeNeDiff, can generalize to datasets without constraints on the format of neural data, while the behavioral data is in the format of timely-aligned video frames. Dataset with these properties are readily accessible in modern experimental settings. Due to the scientific fact that various brain regions, such as SSp and MOs, exhibit different neural latent trajectories related to behaviors of interest, in the paper, we selected the Wide-Field Calcium Imaging dataset [1] to investigate neural dynamics within each region and to analyze discrepancies across different brain regions. Notably, our VDM module is also compatible with single-region Neuropixels data [2] and multi-region voltage imaging data [3]. We will provide the visualization results for these datasets in the appendix of our revised manuscript. We can also include the results in an anonymized link upon request.

**1.2 Main Contribution**

The main modeling and scientific contribution of BeNeDiff both lies in the video diffusion modeling (VDM) module, which interprets the neural dynamics of each disentangled latent factor in a generative manner. Specifically, the VDM module employs a neural encoder (Eq. (7) of the manuscript) to guide the generation of videos that maximize variance in the neural trajectory of target latent factor while minimizing variance in other factors' trajectories. The generation results clearly interprets the neural dynamics of each latent factor, demonstrating specificity to behaviors of interest (e.g., paw-x-axis movement, jaw movement) and aligning with ground-truth behavioral trajectories. This approach ensures that the generated behavioral videos predominantly reflect the neural dynamics of the target latent factor.

**1.3 Baselines Comparison for Neural LVM module**

The following are our comparisons focusing on behaviorally-relevant latent disentangling of the neural LVM module of BeNeDiff. Based on Table 1 of the manuscript, we have added the following three baseline methods for comparison: SSL [4], pi-VAE [5], and CEBRA [6], in which:

* Semi-Supervised Learning (SSL) [4]: a deep generative modeling method that extends the standard variational bound with behavior labels corresponding to each data point.
* pi-VAE [5]: an identifiable variational auto-encoder conditioned on task variables (e.g., motor observable states) for interpretable neural latent discovery.
* CEBRA [6]: a deep neural encoding method that jointly uses behavioural and neural data with constrastive loss for nonlinear neural dynamics discovery.

Please refer to Table 1 of the attached PDF for the results. We observe that, compared to other behavior-informed baseline methods, the neural LVM module of our proposed BeNeDiff achieves the highest disentanglement performance (MIG) while maintaining a high neural reconstruction rate. As the neural LVM module is not our primary contribution, we did not allocate much text and space for it in the manuscript. We will incorporate these comparisons into the appendix of our revised manuscript.  We will also include the corresponding results of these baselines into Figures 4 and 5 of the manuscript.

**1.4 Ablation Study for Neural LVM module**

Please refer to Table 2 of the attached PDF for the results of the variants of our method, demonstrating the contributions of the behavior-informed loss term and the total-correlation penalty loss term. We observe that both terms improve the disentanglement of the neural subspace while generally maintaining neural reconstruction rate. Meanwhile, the neural reconstruction expectation term and the KL regularizer term are basic components of the variational bound and removing them for an ablation study would result in a loss of mathematical integrity. We will incorporate these studies into the appendix of our revised manuscript.

**1.5 Limitation Discussion**

(1) For the neural latent variable model (LVM) module, there exists a balance between disentangling the neural subspace with behavior semantics and maintaining neural reconstruction performance. For each brain region and session, at this stage, a careful hyper-parameter search is necessary to balance the weight between these two components.

(2) For the generative video diffusion module, we implement the neural encoder (classifier for guidance) in Eq. (7) of the manuscript as a linear regressor for interpretability. This linear assumption can be relaxed later for improved guidance performance.

These points will be included in Section 6, 'Discussion', of the revised manuscript.



Refs:

[1] Single-trial neural dynamics are dominated by richly varied movements. (Musall et al., 2019)

[2] Eight-probe Neuropixels recordings during spontaneous behaviors. (Steinmetz et al., 2019)

[3] Widefield imaging of cortical voltage dynamics with an indicator evolved for one-photon microscopy. (Lu et al., 2023)

[4] Semi-supervised learning with deep generative models. (Kingma et al., 2014)

[5] Learning identifiable and interpretable latent models of neural activity using pi-VAE. (Zhou et al., 2020)

[6] Learnable latent embeddings for joint behavioural and neural analysis. (Schneider et al., 2023)

---

### Decision · Program_Chairs · 2024-09-25

**Decision:**

Accept (poster)

**Comment:**

The paper presents a two stage approach for investigating latent representations derived from neural activity.

First, the authors present a TC-VAE with additional supervision (akin to pi-VAE), which they claim provides additional disentanglement of the feature space. Reviewers agree that the claims made regarding disentanglement are too strong, and **should be tuned down in the revision of the paper**. In particular, the standards of empirical evaluation are below what would be usually expected at NeurIPS on a standardized ML benchmark, and the authors should put care into the wording for the camera-ready version.

Regarding experiments, the additional evaluations made during the rebuttal phase are great (especially adding more methods like pi-VAE and CEBRA, and adding the TC regularization there) and should be added to the main manuscript and discussed in the context of disentanglement. Ideally, more disentanglement metrics beyond MIG should be reported in the tables to get a more holistic impression of the model performance.

Finally, the authors should make it clearer that the system is a combination of multiple potentially independent methods, vs. an end-to-end learnable system where the diffusion model informs the latent space -- this seems to be suggested by wording and e.g. the diagram in Figure 2. Better highlighting that the method of estimated the latents is separate from the "visualization" method would improve the clarity.

I recommend acceptance on the paper conditional on these modifications for the camera-ready revision of the paper.